# ASYMMETRIC MOMENTUM:
# A RETHINKING OF GRADIENT DESCENT

## ABSTRACT

Through theoretical and experimental validation, unlike all existing adaptive methods like Adam which penalize frequently-changing parameters and are only applicable to sparse gradients, we propose the simplest SGD enhanced method, Loss-Controlled Asymmetric Momentum(LCAM). By averaging the loss, we divide training process into different loss phases and using different momentum. It not only can accelerates slow-changing parameters for sparse gradients, similar to adaptive optimizers, but also can choose to accelerates frequently-changing parameters for non-sparse gradients, thus being adaptable to all types of datasets. We reinterpret the machine learning training process through the concepts of weight coupling and weight traction, and experimentally validate that weights have directional specificity, which are correlated with the specificity of the dataset. Thus interestingly, we observe that in non-sparse gradients, frequently-changing parameters should actually be accelerated, which is completely opposite to traditional adaptive perspectives. Compared to traditional SGD with momentum, this algorithm separates the weights without additional computational costs. We use multiple networks for our research, employing the datasets Speech Command, Cifar10 and Cifar100 to test the ability for feature separation and conclude phenomena that are much more important than just accuracy rates. Finally, compared to classic SGD tuning methods, we achieve equal or better test accuracy with half the training iterations.
Our demonstration code is available at https://github.com/hakumaicc/Asymmetric-Momentum-LCAM (Not include Authors Information)

## 1 INTRODUCTION

In machine learning, optimizers implement gradient descent. Adaptive optimizers, represented by Adam(Kingma & Jimmy, 2014), have been in competition with traditional optimizers SGD with momentum. From the initial AdaGrad(Duchi et al., 2014) to Rmsprop(Tieleman & Hinton, 2012), and then to Adam and AmsGrad(Loshchilov & Hutter, 2018), adaptive optimizers have undergone many improvements. However, their applicability remains limited to sparse gradients. While adaptive optimizers can converge quickly on sparse datasets, their performance is still not as good as traditional SGD with momentum on non-sparse datasets. Similarly, although SGD can perform well under appropriate scheduling, its convergence is slower. This means that while adaptive optimizers may require twice the computational effort, SGD also needs twice the number of iterations. Consequently, both types of optimizers still play different roles on the datasets where they are most effective. This has also led to some discussions on switching between different optimizers, such as methods for switching from Adam to SGD(Keskar & Socher, 2017).

The saddle point problem(Dauphin et al., 2014) is the primary factor affecting the effectiveness of convergence in the early to mid-stages and also influences the final convergence location. This paper mainly discusses the saddle point issue. Mathematically, a saddle point can be described as a point in a multivariate function where it acts as a local maximum in certain directions and a local minimum in others. In high-dimensional spaces, due to the vast number of directions, finding a genuine local minimum becomes highly challenging, making saddle points more prevalent. The optimization surface of deep learning models is intricate, comprising many saddle points and flat regions.

**The motivation** for this experiment mainly stems from the fact that adaptive optimizers are only well-suited for sparse datasets, as pointed out in AmsGrad(Loshchilov & Hutter, 2018) and many other studies. Meanwhile the most common mechanism of adaptive optimizers is to penalize frequently changing parameters. The inverse conclusion of this phenomenon is that penalizing infrequently changing parameters could better suit non-sparse datasets. Although the conclusion and its inverse are not equivalent, exploring them is still valuable.

**The contributions** of this paper are as follows:

- We first points out that, compared to adding noise simply, machine learning can divided into different momentum stages through the loss values. Through experiment on WRN on Cifar10 and Cifar100, it has been validated two phases represent entirely different behaviours, and there is independence between the two phase positions.

- Compared to traditional methods of stochastic noise, we allows for directional addition of noise in traditional convolutional networks, achieving better accuracy and significantly reducing the required training epochs. In ViT, we found that the noise enhancement disappears in the fine-tuning of transformers, but we still observed noticeable differences brought by Asymmetric Momentum during training. Here, the intensity of penalizing frequently changing parameters exceeds that of adaptive optimizers, with faster convergence speed, further validating the impact of Asymmetric Momentum on training.

## 2 RELATED WORK

### 2.1 SGD

Stochastic Gradient Descent (SGD)(Robbins & Monro, 1951) is a variant of the gradient descent optimization method for minimizing an objective function that is written as a sum of differentiable functions. Instead of performing computations on the whole dataset, which can be computationally expensive for large datasets, SGD selects a random subset (or a single data point) to compute the gradient of the function. Momentum is introduced to the vanilla SGD to make the updates more stable and to dampen oscillations. It takes into account the past gradients to smooth out the updates.

### 2.2 ADAPTIVE OPTIMIZER

Adaptive optimizers,such as AdaGrad, RMSprop, Adam, AmsGrad and AdamW(Loshchilov & Hutter, 2018), typically accumulate past gradients, with different adaptive optimizers employing varied strategies for accumulating historical gradients. Generally, they use accumulated gradients to penalize parameter changes, thereby dynamically adjusting the learning rate for each weight and facilitating faster model convergence.

### 2.3 NOISE

Although our method differs from noise, it builds upon it for additional enhancement. It's evident that our approach also benefits from the effects of noise. Noise is a widely applied technique, with numerous methods for increasing model generalization through noise addition, such as dropout, Label Smoothing, mixup, Positive-Negative Momentum, and the symbol method in the lion optimizer. This even includes Stochastic Gradient Descent (SGD) itself, which inherently contains a certain level of noise.

## 3 LOSS-CONTROLLED ASYMMETRIC MOMENTUM(LCAM)

In this section, we first introduce the concepts of weight coupling and weight traction to provide a simplified explanation of the training process of machine learning. We discuss the general algorithms of adaptive optimizers and speculate on the limitations of adaptive optimizers, specifically their ability to perform well only in the context of sparse gradients. We propose an improved SGD method, Loss-Controlled Asymmetric Momentum(LCAM), which allows for arbitrary combinations of momentum to accelerate in sparse or non-sparse directions. We implement control methods

through the oscillatory properties of the loss value, and provide details of loss behavior during the training process.

## 3.1 WEIGHT COUPLING

Firstly, we introduce the concept of coupling. Representing the input set as $X$ and the output label as $Y$, although the weight segment is quite complex, we simplify it and represented by multiple weight matrices ($\Theta$). We categorize these weights $\Theta$ into three types, $\Theta_c$, $\Theta_s$ and $\Theta_n$. Among them:

$\Theta_n$ signifies **non-sparse** weights that are easily trained, reflecting more prominent features from the raw data and undergo significant changes during training.

$\Theta_s$ stands for **sparse** weights that are hard to train, capturing features that are harder to extract from the raw data, changing minimally throughout the training process.

$\Theta_c$ represents constant weights that are less likely to be influenced by other parameters. It should be noted that, essentially, there is almost no absolute $\Theta_c$. This implies that most parameters can be expressed as $\Theta_{cn}$ or $\Theta_{cs}$, indicating that even neutral parameters will possess a certain degree of bias.

Thus, we can break it down and represent it as:

$$Y = Model(X; \Theta_{j \in n}, \Theta_{k \in s}, \Theta_{l \in cs}, \Theta_{m \in cn}) \tag{1}$$

**Weight Coupling** phenomena are extremely common in machine learning, particularly in convolutions. This is often manifested as oscillations of the model's loss and accuracy within a fixed range under a constant learning rate. It is not until the learning rate is changed that this coupled state transitions to a new state corresponding to the adjusted learning rate.

For this type of Weight Coupling state, it is evident that it is formed by the mutual coupling of multiple parameters. Regarding this coupling, $R$ describe the relation of coupling around global minima, we define it in maths concept as follows:

$$R(\theta_1, \theta_2, ..., \theta_n) = Minima, \theta_n \in \Theta : (\Theta_n, \Theta_s, \Theta_{cn}, \Theta_{cs}) \tag{2}$$

For simplifying the discussion in the following paper, we reduce it to the coupling of two types of parameters $\Theta_n$ and $\Theta_s$. The conclusions drawn from the simplified model need to be remapped to the context of multi-dimensional parameters. The simplified function is showed as follow:

$$R(\Theta_n, \Theta_s) = Minima, \Theta_n \in \Theta, \Theta_s \in \Theta \tag{3}$$

## 3.2 WEIGHT TRACTION

Let's now re-evaluate our understanding of the machine learning training process and Weight Coupling. In the initial stages of training, since the weights are severely offset, changes in individual weights won't cause significant oscillations in their interrelations. Once the model enters an oscillatory state, given that $\Theta_c$, $X$ and $Y$ essentially remains in a quasi-fixed state, it will only produce a certain level of noise from the stochastic gradient output. Thus, $\Theta_n$ and $\Theta_s$ would essentially be in a coupling state, causing both weights groups to oscillate around the optimal point $Minima$, with a certain fixed value as their center.

**Weight Coupling** results in what we term as **Weight Traction Effect**, where the oscillation amplitude is primarily influenced by the learning rate. As illustrated in the figure.1, we demonstrate the phenomena arising from the mutual traction between $\Theta_n$ and $\Theta_s$. The red represents changes dominated by $\Theta_n$, while the green indicates changes dominated by $\Theta_s$. Let's denote the traction force between the weights as $L_\eta$. We can conceptualize $L_\eta$ as a 'traction rod'. This force is influenced by the learning rate. Evidently, the larger the learning rate, the stronger the traction force. The total **Loss** can be described as:

$$Loss_{Total} = Loss_{(L_c, L_\eta)}, L_\eta = Loss_{(L_n, L_s)} \tag{4}$$

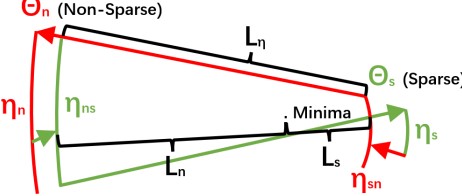

Figure 1: A simple demonstration of the weight coupling state of two groups parameters, $\Theta_n$ and $\Theta_s$.

Therefore, the training process can be understood as being comprised of two distinct parts.

- **Non-Sparse Quick-Changing Weight Group** $\Theta_n$**:** The red part is dominated by $\Theta_n$, where $\Theta_n$ advances by $\eta_n$ through the learning rate. During this phase, $\Theta_n$ exerts a pull on $\Theta_s$ to the left, changing $\Theta_s$'s advance from $\eta_s$ to $\eta_{sn}$.
- **Sparse Slow-Changing Weight Group** $\Theta_s$**:** Similarly, the green part is led by $\Theta_s$, wherein $\Theta_s$ moves forward by $\eta_s$ through the learning rate. In this phase, $\Theta_s$ pulls $\Theta_n$ to the right, causing $\Theta_n$'s progression to shift from $\eta_n$ to $\eta_{ns}$

### 3.3 LIMITATION OF ADAPTIVE OPTIMIZERS

In reality, the inner workings of machine learning are very complex. For simplicity and ease of explanation, we have simplified the training process through the above methods. Next, let's consider the algorithms of adaptive optimizers, and similarly simplify them for ease of discussion. We will start by introducing AdaGrad, followed by a discussion on Adam and fixed-momentum SGD.

For AdaGrad(Duchi et al., 2014), the parameter update rule is:

$$g_t = \nabla J(\Theta_t) \tag{5}$$

$$G_t = G_{t-1} + g_t \odot g_t \tag{6}$$

$$\Theta_{t+1} = \Theta_t - \frac{\eta}{\sqrt{G_t + \epsilon}} \odot g_t \tag{7}$$

$g_t$ is the gradient at time step $t$, $\nabla J$ is the gradient of the objective function $J$ at $\Theta_t$, $G_t$ is a diagonal matrix where each diagonal entry $G_t^{ii}$ is the sum of the squares of the gradients with respect to $\Theta^{ii}$ up to time step $t$. The weights $\Theta$ are updated by subtracting the learning rate $\eta$ divided by the square root of $G_t + \epsilon$ element-wise multiplied by the gradient $g_t$.

As we can see, the core idea behind adaptive optimization is to accumulate historical gradients $G_t$. In the initial stages, it allows for rapid updates of the weights while keeping track of the gradients. In the later stages, the accumulated historical gradients $\sqrt{G_t + \epsilon}$ are used to slow down the update of weights that were updated quickly earlier on, while continuing to train the side of the weights that have been updating more slowly.

To make it more illustrative, we will use the behavior of a **Spring** to analogize the process of machine learning. Just like in Figure.1, we introduce Figure .2 also represents the traction effect between $\Theta_n$ and $\Theta_s$. Consistent with the core idea of adaptive optimization, $\Theta_n$ represents parameters that have historically changed quickly in the non-sparse direction; $\Theta_s$ represents parameters that have historically changed slowly in the sparse direction. The length of the spring represents the learning rate $\eta$. When the learning rate $\eta$ is large, oscillations will be more pronounced; whereas when the spring length is very small, the amplitude of oscillations will also decrease.

The green line represents the path of the adaptive optimizer, which typically penalizes parameters with quickly changes in the non-sparse direction. This is well-suited for sparse datasets as it allows the parameters in the non-sparse direction to quickly settle in the ideal position and slowly trains the parameters in the sparse direction, ultimately leading the parameters to the appropriate destination. However, for non-sparse datasets, it also penalizes the non-sparse direction parameters, this seems

to allow for rapid training, but due to the excessive initial penalization of the non-sparse direction parameters, it ultimately fails to reach the ideal destination.

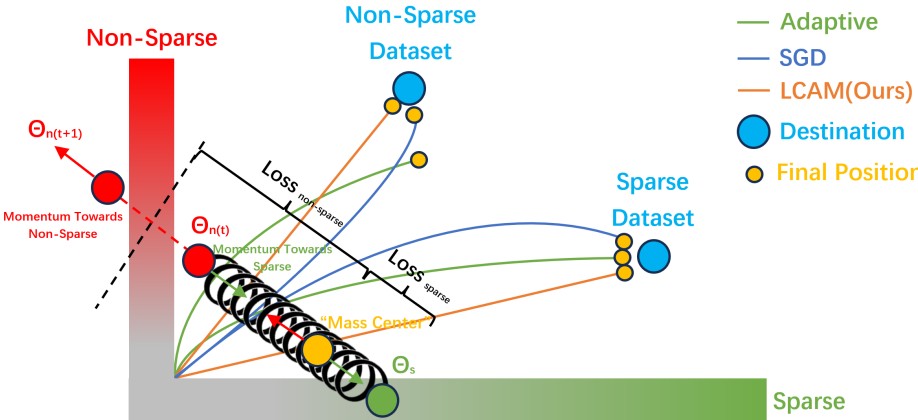

Figure 2: The figure shows the theoretical basis of LCAM, as well as the convergence paths of different optimizers on various datasets. The vertical direction represents the non-sparse direction, which is the direction penalized by adaptive methods, while the horizontal direction represents the sparse direction, which is the primary focus of training in adaptive methods.

Currently, almost all research points to the inability of adaptive methods to converge on non-sparse datasets, and there is also no evidence to suggest that penalizing non-sparse parameters is always effective. Even new optimization methods such as AmsGrad have attenuated the aggressiveness of Adam to achieve better performance. This implies that adaptive optimizers like Adam have a fundamental issue in their mechanism. **Experimentally validating the directionality of parameters and momentum will strongly explain the limitations of adaptive methods.**

Regarding the blue line, which is SGD, it always has better generalization because it can turn slowly, thereby approaching different types of endpoints. However, turning, or actually crossing saddle points, means that it requires more time. The orange line, which is our LCAM, aims to appropriately accelerate at the right stages, thereby finding the right path to speed up SGD.

### 3.4 ASYMMETRIC MOMENTUM

To address the issue of universally destination range applicable, we propose Loss-Controlled Asymmetric Momentum(LCAM). By averaging the loss, we divide training process into different loss phases and using different momentum. We can isolate weights without additional computational cost compared to traditional SGD with momentum. This not only allows for accelerating weights that change slowly in the sparse gradients but also weights that change frequently in non-sparse gradients, thereby making it adaptable to all kinds of datasets.

Considering $\Theta_n$ in Figure.2, in equations (2), we reveal that the loss is composed of three parts, among which $L_c$ is constant. Taking into account the spring system, $\Theta_s$ doesn't change much, so the corresponding $L_s$ also changes very little. Therefore, the apparent changing $Loss_{Total}$ is mostly provided by $L_n$ with a direct ratio:

$$\Delta Loss_{Total} \approx k \times \Delta L_n \tag{8}$$

This means that for the lighter side of the spring, $\Theta_n$, its position can be easily determined by comparing the value of the loss with the average loss. Specifically, when the loss is larger, it will be on the left side of the loss phase line; when the loss is smaller, it will be on the right side of the loss phase line, showed in Figure.2.

Now consider the effect of increasing the momentum at different phases. If we increase the momentum whenever $\Theta_n$ moves to the left side, it's clear that, with the accumulation of applied force, we

will ultimately pull the mass center of the spring system towards the left, which is the red direction of non-sparse gradients. Conversely, if we increase the momentum whenever $\Theta_n$ moves to the right side, then as the force accumulates, we will eventually push the mass center of the spring system towards the right, which corresponds to the green direction of sparse gradients.

$$\text{momentum} = \begin{cases} a & \text{if loss} > \overline{\text{loss}} \\ b & \text{if loss} \leq \overline{\text{loss}} \end{cases} \tag{9}$$

$$\Delta\Theta = -\text{momentum} \times \nabla_{\Theta}\text{loss} \tag{10}$$

$$\Theta_{\text{new}} = \Theta + \Delta\Theta \tag{11}$$

The update strategy is very straightforward. During each iteration, the algorithm compares the current loss with the average loss to determine the current position of $\Theta_n$, thereby deciding whether to apply additional momentum.

### 3.5 LIMITATION

The primary goal of this paper is to explain the limitations of adaptive optimizers. Although the method presented in this paper can be applied in actual training, it still faces some issues. A part of the accuracy improvement in this study originates from noise, which means models that are insensitive to noise or even those for which noise could be destructive may not see an improvement in accuracy. Additionally, our method is more targeted at datasets where parameter directions are asymmetric, implying that for datasets where parameters are more balanced, our method does not offer a performance improvement over methods that only use noise.

## 4 EXPERIMENT

The experiment will be divided into three parts. In the first part, we will use the WRN28-10 model with the Cifar10/100 dataset to demonstrate that a higher momentum of 0.95, compared to the baseline momentum of 0.9, can be divided into two stages based on the loss value. These two stages are asymmetric but can be combined to form a higher momentum to validate our theory on Asymmetric Momentum. In the second part, we test the effect of asymmetric momentum on the ViT16-b model to observe the behaviour of Asymmetric Momentum in transformer networks. In the third part, we introduce the Speech Command dataset and M11 network to test the impact of Asymmetric Momentum in sound recognition, and listed all results of our experiment.

### 4.1 DIRECTION SPECIFICITY DISCUSSION

We designed a very simple experiment to demonstrate the direction specificity of gradients. We tested using WRN28-10(Zagoruyko & Komodakis, 2016), which is from original Residual Network(Kaiming et al., 2016), with the classic Cifar10 and Cifar100(Krizhevsky et al., 2009) as test datasets. Both datasets have 50,000 training samples, but one has ten categories while the other has a hundred. This implies that, in terms of data sample structure, the sparsity level of Cifar100 is ten times that of Cifar10. All experiments are based on SGD, and no dropout was used during testing. Hyperparameters were set with an initial learning rate of 0.1, momentum of 0.9, and weight decay set to $5 \times 10^{-4}$, with a batch size of 128. To more easily test the impact of varying momentum on training, we shortened the drop nodes to epochs 30, 60, and 90, reducing the learning rate to 20% of its value at these epochs. The rapid decline in learning rate deliberately traps the weights in a saddle point, serving as a baseline for discussing asymmetric momentum.

#### 4.1.1 CIFAR10

First, we conducted experiments on Cifar10. We used a total of four groups of momentum:

- The first and second groups maintain a momentum of 0.9 and 0.95 throughout, serving as the baseline. These are represented by black and blue lines, respectively.

- The third group uses a momentum of 0.95 during the sparse phase on the right and 0.9 during the non-sparse phase on the left, to accelerate the sparse $\Theta_s$ side and push the weights to the right. This is represented by a green line.

- The fourth group uses a momentum of 0.95 during the non-sparse phase on the left and 0.9 during the sparse phase on the right, to accelerate the non-sparse $\Theta_n$ side and push the weights to the left. This is represented by a red line.

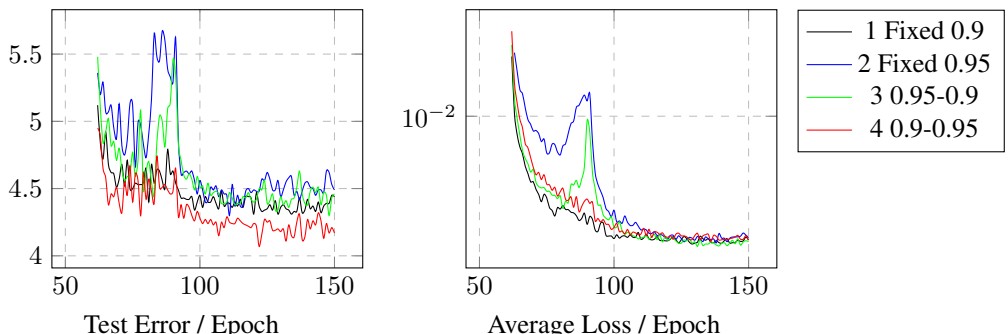

Figure 3: Asymmetric Momentum tested on WRN28-10 with Cifar10. The fourth group with the asymmetric momentum achieved the best performance.

In this experimental setup in Figure.3, the impact brought by the distinct direction specificity is very evident. We can observe that the second group, which maintains a momentum of 0.95 throughout, has an excessive momentum that prevents it from making timely turns before reaching the destination, thus missing the optimal point. Compared to the first group, there is a significant abnormal increase in both its error rate and loss value.

In the third group, we observed a similar phenomenon. Compared to the fixed 0.95 momentum, its abnormal increase occurred later. This is because, even though we were continuously pushing to the right, or the sparse direction, the bias should be more. However, due to the speed of descending along the gradient descent, which is slower compared to the continuous 0.95 momentum in the second group, it caused the timing of the abnormality to be later than the fixed 0.95 momentum group.

Looking further into the fourth group, **only swap momentum directions** in previous group, which uses a momentum of 0.95 during the non-sparse phase, it's evident that no anomalies occurred, and its final performance is significantly better than the other groups. This clearly demonstrates that the properties of $\Theta_n$ and $\Theta_s$ are distinct, indicating that the weights are specificity in nature. Furthermore, it also shows that Cifar10 is a non-sparse dataset, consistent with the composition of datasets.

### 4.1.2  CIFAR100

In the Cifar100 dataset, the situation underwent a noticeable change. Even though we set up the same experimental combinations, the results were distinctly different from those in Cifar10. The four groups are same as Cifar10.

In the first and second baseline group, unlike in Cifar10 where it's quite evident, we can essentially consider both fixed momentum training to be without anomalies lines. However, from the training results, it's easy to see that the outcome with a momentum of 0.95 is not ideal. This is because the weights changes too quickly, causing SGD to be unable to adjust its direction in time.

In the third group, we obtained the best results. We can deduce that in Cifar100, pushing the weights towards the sparse side yields better outcomes, implying that Cifar100 is a sparse dataset, consistent with the structure of the dataset.

However, after **only swap momentum directions**, we noticed that the fourth group experienced oscillations in the mid-training phase. This result is the most confused. When we intentionally pushed $\Theta_n$ to the left, it was pushed too far, causing a significant oscillation in the overall weights.

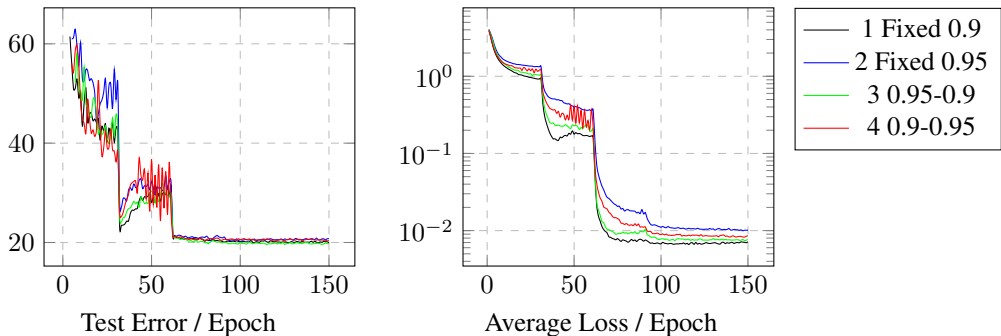

Figure 4: Asymmetric Momentum tested on WRN28-10 with Cifar100. We still only made a swap in the momentum direction, yet it resulted in different effects compared to Cifar10.

This is the tug-of-war between the traction produced by the learning rate in the spring model and the pushing force generated by the Asymmetric Momentum. This further verifies the correctness of our theory.

## 4.2 VISION TRANSFORMER

Discussing the application of an optimization method in ViT (Vision Transformer) is highly valuable. As one of the most popular methods currently, using ViT's pre-trained models for fine-tune can achieve excellent accuracy in a short time. We chose the pre-trained model of ViT-16B on ImageNet21k for fine-tune on Cifar10 and Cifar100.

Since the experiment no longer needs to test excessively high momentum and is more oriented towards practical application, we set the learning rate in SGD to 1e-4, with momentum shifts roughly around 0.9, 0.857-0.93, and 0.93-0.857. We kept the momentum as close to 0.9 as possible to minimize the shift in average momentum. Due to iterative effects, the Asymmetric Momentum Value(a-b) should satisfy $(1 - 0.9)^2 = (1 - a) * (1 - b)$; when one side is set to 0.93, the other should be approximately 0.857 to maintain average momentum.

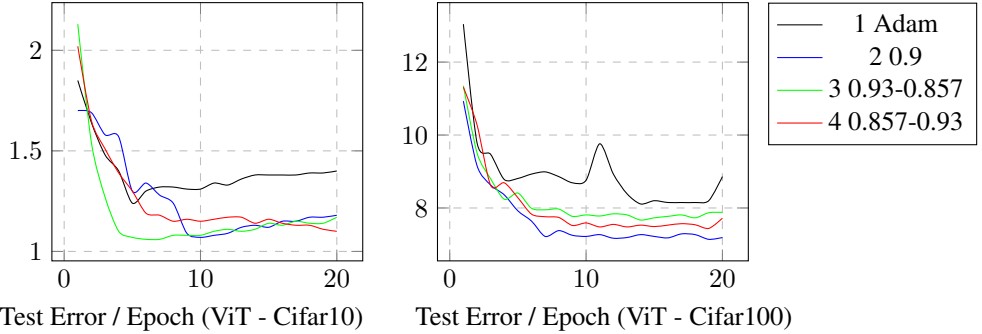

Figure 5: Asymmetric Momentum tested on Vit-16B.

The results of the experiment, as shown in the Figure.5, indicate that Asymmetric Momentum performs differently in Cifar10 and Cifar100.

In Cifar10, we can see that all three set of hyper-parameters can achieve optimal accuracy point, but it's clear that the green line, which mimics the adaptive method to accelerate the sparse direction, reaches the endpoint the fastest, but subsequently exhibits over-fitting. Similarly, the default state of SGD, after reaching the optimal point, also shows over-fitting. The red line, which accelerates in the non-sparse direction, experiences no over-fitting at the latest. This suggests that Asymmetric Momentum can significantly impacts the model's convergence behaviour.

However, in Cifar100, we did not observe any improvement in training due to Asymmetric Momentum, including noise. This means that when fine-tune Cifar100 with ViT16-B, the direction

specificity of its parameters is minimal, and we do not need complex tricks to achieve the desired accuracy.

### 4.3 OTHER TEST AND FINAL RESULTS

We additionally tested the M11 network on the Speech Command dataset, applying the same three momentum strategies for comparison. All results have been compiled and are presented in the table as shown.

Table 1: Final results of full experiments.

| Dateset | Network | - | Scheduler | M0.9 | M0.857-0.93 | M0.93-0.857 |
|---|---|---|---|---|---|---|
| Speech Command | M11 | Ours | SGD+Cdlr | $5.788_{0.142}$ | $\mathbf{5.470}_{0.091}$ | $5.625_{0.173}$ |
| Cifar10 | WRN2810 | Ours | SGD+Cdlr | $4.11_{0.09}$ | $\mathbf{3.95}_{0.12}$ | $4.05_{0.08}$ |
| | ViT-B16 | Google | SGD+Cos | 1.09 | - | - |
| | | - | Adam | $1.58_{0.37}$ | - | - |
| | | Ours | SGD | $1.24_{0.07}$ | $\mathbf{1.20}_{0.08}$ | $1.25_{0.07}$ |
| Cifar100 | WRN2810 | Ours | SGD+Cdlr | $19.21_{0.08}$ | $19.24_{0.13}$ | $\mathbf{19.11}_{0.21}$ |
| | ViT-B16 | Google | SGD+Cos | 7.94 | - | - |
| | | - | Adam | $8.00_{0.63}$ | - | - |
| | | Ours | SGD | $\mathbf{7.37}_{0.18}$ | $7.61_{0.12}$ | $7.59_{0.24}$ |

## 5 CONCLUSION

Although the experiment is simple, the phenomena are obvious. We explain why adaptive optimizers only suit for sparse gradient, prove that within the model weights, the gradient has specificity on various directions based on the dataset sparsity. We introduced the theories of Weight Coupling and Weight Traction and utilized this mechanism through Asymmetric Momentum. By comparing the current loss value with the average loss value after each training iteration, we segmented the training process into two phases with different momentum towards sparse or non-sparse direction, and can be adapted for different datasets. In training, Loss-Controlled Asymmetric Momentum(LCAM) retains the benefits of traditional SGD, achieving the best accuracy with minimal training resources required for each iteration. Additionally, the demand epochs for training can be nearly halved.

## 6 REPRODUCIBILITY STATEMENT

Figure.3 and Figure.4 are our main experiments, validating the correctness of our theory. The abnormal phenomena in the specially designed experiment is merely affected by the local minima, leading to extremely high experimental stability and 100% Reproducibility.

However, it needs to be stated that, in Figure.5, the final test error is significantly affected by the local minima, which is not the part we focus on. After extensive experimentation, compared with a fixed momentum of 0.9, the probability of improvement by our method is roughly 80% in Cifar10, 50% in Cifar100. This implies that choosing a more appropriate scheduler to replace our simple one could potentially yield better stability.

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
