# OpenReview forum: "Asymmetric Momentum: A Rethinking of Gradient Descent"
_ICLR.cc/2024/Conference — Submitted to ICLR 2024_

### Official Review · Reviewer_SnV5 · 2023-10-30

**Soundness:** 1 poor
**Presentation:** 2 fair
**Contribution:** 1 poor
**Rating:** 1
**Confidence:** 4

**Summary:**

In this paper, a new strategy of setting momentum is proposed, called loss-controlled asymmetric momentum (LCAM). The aim is to make the momentum adaptable to different tasks. The method is based on heuristic observation and is evaluated numerically.

**Strengths:**

As an important technique used in neural network training, momentum is indeed important. Discussion and effort on improving its performance is encouraged.

**Weaknesses:**

In this paper, the whole discussion about momentum is heuristic not rigorous. Indeed, the setting of momentum in the existing strategy is far from ideal, however, it is very hard to find a simple rule, as done in this paper, to determine it.

Since the discussion is not convincing, the authors have to use experiments to show the advantage of the proposed methods. However, the experiments are not convincing neither. To show the advantages over popular optimizers, the experiments should include different structures (CNN/ViT/w/o BN/w/o skip connection, etc.), different tasks (imagenet, segmentation, detection, etc.), different scenario (different initializations, different setting), and different baselines (different optimizers, different setting, and different recent modifications). Most importantly, the setting of other methods should be good, e.g., using some well-accepted setting. Overall, the current experiments are not sufficient: one can always cherry-pick good result for a heuristic strategy.

**Questions:**

please see the weakness for numerical experiments. I do expect to see additional and more convincing results. Maybe the time is not sufficient for ICLR2024, but hope later I could see the proposed method in other conference.

---

> ### Author Response · Authors · 2023-11-23
>
> Due to the limited time, the supplementary experiments did not cover a wider range of datasets, and there was not enough time to revise the paper. I have uploaded this version in the hope of receiving further guidance and comment. More content will be added in the future.
>
> 1、I have included experiments on asymmetric momentum in Vit-16_B, including a comparison with Adam.
> 2、To improve the logic, we added motivation and revised the explanation of Figure 2 to more vividly describe the effect of asymmetric momentum.
> 3、Some formulas were modified to reflect the actual situation.
> 4、Added experiments on sound recognition.
>
> I am aware of some issues that could not be addressed due to time constraints, including:
>
> 1、Although we have demonstrated the effect of asymmetric momentum through logical theoretical methods and experiments, achieving a logical loop and explaining the limitations of adaptive methods, we believe this is not a heuristic paper. We have provided several experiments that prove the impact of asymmetric momentum on model training, and the supplementary experiments also support the theory. However, we have not been successful in finding a formula to fully explain the theory.
> 2、Due to time constraints, although some methods and related research are mentioned in the supplement, there was not enough time to add references, and some of the writing may be problematic.
> 3、Also due to time constraints, complete testing on some datasets, such as ImageNet and object detection, could not be performed.
> If there are more suggestions, I would appreciate them.

---

### Official Review · Reviewer_Erbu · 2023-10-30

**Soundness:** 2 fair
**Presentation:** 2 fair
**Contribution:** 2 fair
**Rating:** 5
**Confidence:** 4

**Summary:**

The paper introduces a novel method called Loss-Controlled Asymmetric Momentum (LCAM) to enhance the Stochastic Gradient Descent (SGD) optimization process. Unlike existing adaptive methods such as Adam, which are primarily suitable for sparse gradients, LCAM is designed to be adaptable to all types of datasets. The authors propose averaging the loss to segment the training process into different phases, each with its distinct momentum. The paper also introduces the concepts of weight coupling and weight traction, suggesting that weights have a directional specificity based on dataset sparsity. The experiments primarily utilize Wide Residual Networks (WRN) on the Cifar10 and Cifar100 datasets. The results indicate that LCAM can achieve comparable or better accuracy with nearly half the training epochs compared to traditional SGD methods.

**Strengths:**

1. The introduction of LCAM provides a fresh perspective on optimizing the gradient descent process, especially in the context of non-sparse gradients.
2. The paper provides a solid theoretical foundation, introducing concepts like weight coupling and weight traction.
3. The experiments on Cifar10 and Cifar100 using WRN provide empirical evidence supporting the proposed method's efficacy.
4. The authors emphasize the reproducibility of their experiments, which is crucial for the scientific community to validate and build upon their findings.

**Weaknesses:**

1. The paper delves deep into theoretical aspects, which might make it challenging for readers unfamiliar with the topic.
2. The experiments are primarily conducted on Cifar10 and Cifar100. Testing on a broader range of datasets would provide a more comprehensive understanding of LCAM's applicability.
3. The mechanism for reducing the learning rate at every iteration is based on empirical observations. A more systematic approach or justification would strengthen the paper's claims.
4. The influence of local minima on the final test error is acknowledged but not deeply explored, which might leave some questions unanswered for the readers.

**Questions:**

Please see weaknesses.

---

> ### Author Response · Authors · 2023-11-23
>
> Due to the limited time, the supplementary experiments did not cover a wider range of datasets, and there was not enough time to revise the paper. I have uploaded this version in the hope of receiving further guidance and comment. More content will be added in the future.
>
> 1、I have included experiments on asymmetric momentum in Vit-16_B, including a comparison with Adam.
> 2、To improve the logic, we added motivation and revised the explanation of Figure 2 to more vividly describe the effect of asymmetric momentum.
> 3、Some formulas were modified to reflect the actual situation.
> 4、Added experiments on sound recognition.
>
> I am aware of some issues that could not be addressed due to time constraints, including:
>
> 1、Although we have demonstrated the effect of asymmetric momentum through logical theoretical methods and experiments, achieving a logical loop and explaining the limitations of adaptive methods, we believe this is not a heuristic paper. We have provided several experiments that prove the impact of asymmetric momentum on model training, and the supplementary experiments also support the theory. However, we have not been successful in finding a formula to fully explain the theory.
> 2、Due to time constraints, although some methods and related research are mentioned in the supplement, there was not enough time to add references, and some of the writing may be problematic.
> 3、Also due to time constraints, complete testing on some datasets, such as ImageNet and object detection, could not be performed.
> If there are more suggestions, I would appreciate them.

---

### Official Review · Reviewer_KDdh · 2023-11-01

**Soundness:** 2 fair
**Presentation:** 2 fair
**Contribution:** 3 good
**Rating:** 3
**Confidence:** 3

**Summary:**

This paper proposes a framework to understand the effects of data sparsity on different optimizers. To this end, it separates weights into non-sparse and sparse groups that change quickly and slowly during training, respectively. Then it proposes a weight-traction model to justify the underperformance of adaptive methods (such as Adagrad or Adam) on non-sparse dataset (e.g. CIFAR10). The main argument is that the rapid change in the non-sparse weights (caused by rapid decrease in the corresponding learning rates) causes the overall weight shifting towards the sparse side. To accommodate datasets of different sparsity, this works proposes a method that uses different momentum parameters for sparse and non-sparse training phase, which is determined by comparing the current loss to the average loss.  It empirically verifies that choosing a proper momentum parameter for non-sparse or sparse weights (depending on dataset sparsity) leads to better performance.

**Strengths:**

- Some interesting experimental observations are reported. Specifically, Figure 3 and Figure 4 show that accelerating different parameter groups (sparse or non-sparse depending on the nature of the dataset) seems to lead to better test error.

- The determination of sparse or non-sparse phase based on the loss seems to be intuitive given the non-sparse weights change more frequently and contribute more to the overall loss change.

**Weaknesses:**

- The justifications and the framework are purely heuristic. There is no quantitative arguments or actual theory to concretely explain the observed phenomenon. The linear model (e.g. eqn 1) is overly simplified and may not be able to capture the training dynamics of a non-linear neural network.

- The proposed algorithm is rather restrictive to the models that are (such as wide residual network) able to extract features, which limits its applicability in other scenarios.

- The current related work section is not informative and missing a lot of references. More background on the training dynamics of momentum and comparisons of SGD and Adam on various tasks are required.

**Questions:**

N/A

---

> ### Author Response · Authors · 2023-11-23
>
> Due to the limited time, the supplementary experiments did not cover a wider range of datasets, and there was not enough time to revise the paper. I have uploaded this version in the hope of receiving further guidance and comment. More content will be added in the future.
>
> 1、I have included experiments on asymmetric momentum in Vit-16_B, including a comparison with Adam.
> 2、To improve the logic, we added motivation and revised the explanation of Figure 2 to more vividly describe the effect of asymmetric momentum.
> 3、Some formulas were modified to reflect the actual situation.
> 4、Added experiments on sound recognition.
>
> I am aware of some issues that could not be addressed due to time constraints, including:
>
> 1、Although we have demonstrated the effect of asymmetric momentum through logical theoretical methods and experiments, achieving a logical loop and explaining the limitations of adaptive methods, we believe this is not a heuristic paper. We have provided several experiments that prove the impact of asymmetric momentum on model training, and the supplementary experiments also support the theory. However, we have not been successful in finding a formula to fully explain the theory.
> 2、Due to time constraints, although some methods and related research are mentioned in the supplement, there was not enough time to add references, and some of the writing may be problematic.
> 3、Also due to time constraints, complete testing on some datasets, such as ImageNet and object detection, could not be performed.
> If there are more suggestions, I would appreciate them.

---

### Official Review · Reviewer_cZTD · 2023-11-01

**Soundness:** 1 poor
**Presentation:** 2 fair
**Contribution:** 1 poor
**Rating:** 3
**Confidence:** 4

**Summary:**

The paper introduces a variant to SGD, named Loss-Controlled Asymmetric Momentum (LCAM), aiming to adaptively accelerate both slow-changing parameters for sparse gradients and frequently-changing parameters for non-sparse gradients. The method divides the training process into different loss phases, utilizing different momentum values accordingly.

**Strengths:**

The authors make an effort to explain the proposed method in an intuitive way.

**Weaknesses:**

1. Despite the attempt to give an intuitive explanation, many of the concepts are not well defined or explained, e.g., weight coupling, oscillatory state, coupling state. Overall, section 3 is difficult to follow, and the motivation is not convincing.
2. The experiments are only conducted on CIFAR10/100 with wide resnet, and do not show significant improvement. Moreover, the accuracy values do not have confidence intervals.
3. It is unclear how the multiple hyperparameters are determined, and no ablation study is provided to justify the design choices.
4. Some of the experimental results seem inconsistent. For instance, curves 1 and 4 in Fig. 4 do not match at the early stage of training when they share the same momentum value.

**Questions:**

See above.

---

> ### Author Response · Authors · 2023-11-23
>
> Due to the limited time, the supplementary experiments did not cover a wider range of datasets, and there was not enough time to revise the paper. I have uploaded this version in the hope of receiving further guidance and comment. More content will be added in the future.
>
> 1、I have included experiments on asymmetric momentum in Vit-16_B, including a comparison with Adam.
> 2、To improve the logic, we added motivation and revised the explanation of Figure 2 to more vividly describe the effect of asymmetric momentum.
> 3、Some formulas were modified to reflect the actual situation.
> 4、Added experiments on sound recognition.
>
> I am aware of some issues that could not be addressed due to time constraints, including:
>
> 1、Although we have demonstrated the effect of asymmetric momentum through logical theoretical methods and experiments, achieving a logical loop and explaining the limitations of adaptive methods, we believe this is not a heuristic paper. We have provided several experiments that prove the impact of asymmetric momentum on model training, and the supplementary experiments also support the theory. However, we have not been successful in finding a formula to fully explain the theory.
> 2、Due to time constraints, although some methods and related research are mentioned in the supplement, there was not enough time to add references, and some of the writing may be problematic.
> 3、Also due to time constraints, complete testing on some datasets, such as ImageNet and object detection, could not be performed.
> If there are more suggestions, I would appreciate them.

---

### Public Comment · ~Zeke_Xie1 · 2023-11-19
**Related Work on Momentum Methods**

Dear Authors,

Momentum is a class of very important methods for stochastic optimization and deep learning.

However, I believe it is very necessary to review related works on revising/improving momentum methods in such a paper.

For example, two of my recent works on momentum methods are very relevant to this topic.


Reference:

[1] Xie, Z., Yuan, L., Zhu, Z., & Sugiyama, M. (2021, July). Positive-negative momentum: Manipulating stochastic gradient noise to improve generalization. In International Conference on Machine Learning (pp. 11448-11458). PMLR.

[2] Xie, Z., Wang, X., Zhang, H., Sato, I., & Sugiyama, M. (2022, June). Adaptive inertia: Disentangling the effects of adaptive learning rate and momentum. In International conference on machine learning (pp. 24430-24459). PMLR.

---

### Meta-Review · Area_Chair_nNNB · 2023-12-05

**Metareview:**

This paper  introduces  Loss-Controlled Asymmetric Momentum (LCAM) to improve SGD. LCAM averages the loss to segment the training process into different phases, each with its distinct momentum. It  also introduces the concepts of weight coupling and weight traction, and shows that weights have a directional specificity based on dataset sparsity. Experimental results on Wide Residual Networks (WRN) on the Cifar10 and Cifar100 datasets shows some effectiveness.

The main strength of this work is that its learning framework is a theory-sound framework to better learn different frequencies in the diffusion model. However, many reviewers emphasize 1) the unclear writing and definitions (e.g., weight coupling, oscillatory state) which leads to unclear and convincing motivation,  2) insufficient experiments, such as no experiments on large-scale datasets and networks, unclear setting of multiple hyperparameters, 3) analysis on oversimplified linear model which may not be extended to network.  I partly agree with the reviewer's comments, especially for the writing and experiments. So to improve this work, the authors can revise it accordingly.

Since most reviewers have negative feedback, we cannot accept it in this conference.

**Justification For Why Not Higher Score:**

many reviewers emphasize
1) the unclear writing and definitions (e.g., weight coupling, oscillatory state) which leads to unclear and convincing motivation,

2) insufficient experiments, such as no experiments on large-scale datasets and networks, unclear setting of multiple hyperparameters,

3) analysis on oversimplified linear model which may not be extended to network.

 I partly agree with the reviewer's comments, especially for the writing and experiments.

**Justification For Why Not Lower Score:**

N/A

---

### Decision · Program_Chairs · 2024-01-16

Reject